# Transcriptome Analysis of Sweet Cherry (*Prunus avium* L.) Cultivar ‘Lapins’ upon Infection of *Pseudomonas syringae* pv. *syringae*

**DOI:** 10.3390/plants12213718

**Published:** 2023-10-29

**Authors:** Weier Cui, Nicola Fiore, Franco Figueroa, Carlos Rubilar, Lorena Pizarro, Manuel Pinto, Set Pérez, María Francisca Beltrán, Claudia Carreras, Paula Pimentel, Alan Zamorano

**Affiliations:** 1Laboratorio de Fitovirología, Departamento de Sanidad Vegetal, Facultad de Ciencias Agronómicas, Universidad de Chile, Santiago 8820808, Chile; cuiweierpku@gmail.com (W.C.); nfiore@uchile.cl (N.F.); fran.ibv24@gmail.com (M.F.B.); claudia.carreras@gmail.com (C.C.); 2Laboratorio de Inmunidad Vegetal, Instituto de Ciencias Agroalimentarias, Universidad de O’Higgins, San Fernando 3070000, Chile; franco.figueroa@uoh.cl (F.F.); carlos.rubilar@uoh.cl (C.R.); lorena.pizarro@uoh.cl (L.P.); manuel.pinto@uoh.cl (M.P.); 3Laboratorio de Patología Vegetal, Instituto de Ciencias Agroalimentarias, Animales y Ambientales, Universidad de O’Higgins, San Fernando 3070000, Chile; set.perez@uoh.cl; 4Laboratorio de Fisiología del Estrés, Centro de Estudios Avanzados en Fruticultura (CEAF), Camino Las Parcelas, 882, Rengo 2940000, Chile; ppimentel@ceaf.cl

**Keywords:** bacterial canker, differentially expressed genes, GO enrichment, disease-related genes, local responses, distal responses

## Abstract

Bacterial canker caused by *Pseudomonas syringae* pv. *syringae* (Pss) is responsible for substantial loss to the production of sweet cherry in Chile. To date, the molecular mechanisms of the Pss–sweet cherry interaction and the disease-related genes in the plant are poorly understood. In order to gain insight into these aspects, a transcriptomic analysis of the sweet cherry cultivar ‘Lapins’ for differentially expressed genes (DEGs) in response to Pss inoculation was conducted. Three Pss strains, A1M3, A1M197, and 11116_b1, were inoculated in young twigs, and RNA was extracted from tissue samples at the inoculation site and distal sections. RNA sequencing and transcriptomic expression analysis revealed that the three strains induced different patterns of responses in local and distal tissues. In the local tissues, A1M3 triggered a much more extensive response than the other two strains, enriching DEGs especially involved in photosynthesis. In the distal tissues, the three strains triggered a comparable extent of responses, among which 11116_b1 induced a group of DEGs involved in defense responses. Furthermore, tissues from various inoculations exhibited an enrichment of DEGs related to carbohydrate metabolism, terpene metabolism, and cell wall biogenesis. This study opened doors to future research on the Pss–sweet cherry interaction, immunity responses, and disease control.

## 1. Introduction

Sweet cherry (*Prunus avium* L.) ranks first in fruit tree cultivation and fruit production in Chile [1]. During the 2021–2022 season, a total of 259,202.5 tons of fruit were exported to the international market, making Chile the top exporting country in the world [2]. With the increasing cultivation, diseases have emerged, devastating the crop trees and impairing fruit production. Among these diseases, bacterial canker has become a main cause of yield loss [3]. The pathovars *P. syringae* pv. *syringae* and *P. amygdali* pv. *morsprunorum* have been isolated from the symptomatic trees and confirmed to be the causal pathogens [4,5]. Between them, *P. syringae* pv. *syringae* (Pss) is the major cause of the disease, resulting in up to 40% of tree losses [6].

To control the disease and minimize yield loss, studies have been conducted to comprehend the interactions between the pathogen and the host during the development of bacterial canker disease. So far, most research in this field has focused on the pathogenicity of *P. syringae* pvs., leaving a significant knowledge gap concerning disease-related genes in the host. A similar lack of knowledge is also evident in other *Prunus* species affected by bacterial canker, even though phenotype screenings have identified cultivars displaying differential susceptibilities to the disease [7,8,9,10]. In a recent study involving apricot (*P. armeniaca* L.), genome-wide single-nucleotide polymorphism markers revealed associations between seven chromosomal loci and plant resistance to Pss [11]. However, no candidate genes for resistance to bacterial canker were identified within these loci.

The identification of disease-related genes in sweet cherry necessitates whole-genome sequencing and gene annotation. Currently, GenBank provides access to the genomes of four *P. avium* cultivars: ‘Satonishiki’, the leading variety in Japan [12]; ‘Big Star’, a self-crossing progeny of ‘Lapins’ generated by the University of Bologna [13,14]; ‘Tieton’, a hybrid of ‘Stella’ and ‘Early Burlat’ created by Washington State University [15]; and ‘Hongmanao’, the most extensively cultivated sweet cherry variety in north China (GenBank BioProject PRJNA489346). Gene prediction and annotation have been performed on ‘Satonishiki’ and ‘Tieton’, the former being available in GenBank. Additionally, a basic transcriptome analysis of three cultivars, ‘Bing’, ‘Lapins’ and ‘Rainier’, has been conducted using the annotation data [16].

This study presents the first comparative transcriptome analysis of sweet cherry infected with *P. syringae* pv. *syringae* (Pss). Three Pss strains of varying virulence were used to inoculate sweet cherry through wounding, and RNA was extracted from the tissue near and away from the wound. RNA sequencing revealed differentially expressed genes post-infection with each strain, as well as the genes induced by two or three individual strains. Gene ontology enrichment analysis unveiled unique and shared responses elicited by the three strains. These datasets address the scarcity of disease-related gene candidates in sweet cherry and open the way for extensive research into the defense responses of sweet cherry against this bacterium.

## 2. Results

### 2.1. Disease Development

Three Pss strains, A1M3, A1M197, and 11116_b1, were inoculated into twigs of sweet cherry cv. ‘Lapins’. Forty days after Pss inoculation, wounds with either an opening or filled irregular cork structure with gum, typical of bacterial canker, were observed for all three strains. By contrast, the mock-treated twigs displayed a closed wound covered with compact cork, giving it a clean appearance (Appendix A). Lesions induced by Pss were observed up to 5 cm around the inoculation zone. To confirm the effectiveness of Pss infection in the twigs, the presence of viable Pss in the local samples near the wound was assessed (Appendix A). The colonies analyzed from the infected twigs exhibited UV-induced fluorescence with a LOPAT profile consistent with Pss [17] (Appendix A). DNA was extracted from both local and distal samples, and Pss DNA was quantified using Real-time PCR (Appendix A). In the local samples, Pss was detected in all Pss-inoculated twigs, whereas in the mock-treated samples, only one sample tested positive for Pss, with an amount of bacterial DNA lower than that of the Pss-inoculated samples (Appendix A). The A1M197-inoculated samples exhibited the highest Pss DNA content, nearly 50 times higher than that of the A1M3-inoculated samples. The 11116_b1-inoculated samples presented an amount of bacterial DNA that ranged between the values of the other two samples. Intriguingly, Pss was also detected in one of the three replicates of the distal samples from both A1M197- and A1M3-inoculated twigs (Appendix A). These results confirmed that the three Pss strains were capable of establishing in the cv. ‘Lapins’ twigs around the inoculation zone, and their infection was effective 40 days after inoculation.

### 2.2. The Transcriptome Datasets of Sweet Cherry cv. ‘Lapins’

The RNA sequencing of each of the 24 samples generated 53,215,158 to 69,961,046 raw reads, each with a length of 151 nt (Table 1). Within each dataset, 92.03% to 96.77% of reads reached the quality of Q20, and 86.79% to 92.50% of reads reached the quality of Q30. After quality trimming, 46,916,794 to 66,028,852 trimmed reads were obtained, accounting for 85.89% to 95.40% of the raw reads and resulting in average read lengths of 126.38 to 134.25 nt.

For 22 out of 24 datasets, 21.66% to 39.59% of the trimmed reads were mapped to the reference transcriptome of *Prunus avium* cv. ‘Satonishiki’, covering 67.01% to 75.29% of the total of 35,009 transcripts. Two datasets corresponding to an A1M3-inoculated local sample and an A1M3-inoculated distal sample exhibited significantly lower percentage of mapped reads compared to the others, with values of 6.64% and 9.18%, respectively. However, the low percentage was not related to the quality of the reads.

Principal component analysis (PCA) of the mapped transcripts from all the datasets revealed two clusters with three outliers (Figure 1A, Appendix A). The two clusters included 11 local samples and 10 distal samples, respectively, indicating that they exhibited similar expression profiles for the majority of the transcripts. The three outliers corresponded to the datasets with the shortest average read lengths after trimming and the lowest percentages of reads mapped to the reference transcriptome (Table 1). This observation suggests that the difference in clustering might be attributed to the length and number of mapped reads. In order to further separate the clusters, the outliers were removed, and the remaining 21 datasets were reanalyzed. The new scatter plot revealed that, for the local samples, the datasets clustered distinctly based on different treatments, whereas for the distal samples, the datasets from various treatments did not exhibit significant clustering patterns (Figure 1B). These findings imply that inoculation with different Pss strains resulted in more diverse local responses. However, removing the three outliers did not alter the results of the expression analysis; therefore, they were retained for the subsequent analyses.

### 2.3. Different Pss Strains Trigger Different Local and Distal Responses

Differential expression analysis revealed that the three Pss strains elicited different local and distal responses in sweet cherry (Appendix A). In the local samples, A1M3 inoculation caused the differential expression of 708 genes (Appendix A, Figure 2A). In contrast, A1M197 caused the differential expression of 198 genes, whereas 11116_b1 only caused the differential expression of 85 genes (Appendix A, Figure 2A). This difference indicates that A1M3 triggered a more significant local response in sweet cherry than the other two Pss strains. A1M3 and A1M197 inoculations shared 92 DEGs, accounting for 46.5% of the total DEGs caused by A1M197 but only 13.0% of that of A1M3 (Figure 2A). Similarly, A1M3 and 11116_b1 inoculations shared 37 DEGs, representing 43.5% of the total DEGs caused by 11116_b1 but only 5.2% of that of A1M3 (Figure 2A). In the distal samples, however, inoculation with A1M3, A1M197, and 11116_b1 caused the differential expression of 354, 336, and 427 genes, respectively (Appendix A, Figure 2D), suggesting that the three strains triggered a comparable extent of distal response in the plant.

The ratios of up- and down-regulated genes in response to the inoculation showed different patterns in the local and distal datasets. In the local samples, the numbers of up- and down-regulated genes were comparable: 398 and 310 for A1M3, 73 and 125 for A1M197, and 38 and 47 for 11116_b1, respectively (Figure 2B,C, Appendix A). By contrast, in the distal samples, the number of up-regulated genes was significantly higher than the number of down-regulated genes for all three Pss strains, with the former being two to six times higher than the latter: 297 and 53 for A1M3, 228 and 106 for A1M197, and 337 and 87 for 11116_b1, respectively (Figure 2E,F, Appendix A). These data suggest that the response to the inoculation in the distal tissue could be more specific to disease resistance than in the local tissue.

Four DEGs included in Figure 2D were not presented in Figure 2E,F due to their different responses in different treatments (Appendix A; Figure 3A). XM_021971224.1, which encodes a homolog of an *Arabidopsis* G-type lectin S-receptor-like serine/threonine-protein kinase, was up-regulated by A1M3 inoculation but down-regulated by A1M197 inoculation. XM_021954392.1, which encodes a probable aquaporin, and XM_021974777.1, which encodes a probable L-cysteine desulfhydrase in the chloroplast, were up-regulated by A1M3 inoculation but down-regulated by 11116_b1 inoculation. XM_021953649.1, which encodes a (3S,6E)-nerolidol synthase 1-like protein, was up-regulated by A1M3 inoculation but down-regulated by A1M197 and 11116_b1 inoculation.

The fold change in genes also varied among the inoculations with the three Pss strains. In the local samples from A1M3 inoculation, DEGs that were regulated by 2- to 16-fold constituted over 80% of all DEGs, whereas the percentages for A1M197 and 11116_b1 were significantly lower (Appendix A). This suggests that the local response triggered by A1M3 was more extensive, rather than stronger, than that of the other two strains. On the other hand, genes up- or down-regulated more than 256-fold accounted for a small percentage of the total DEGs from all three inoculations (8.3%, 7.6%, and 5.9% for A1M3, A1M197, and 11116_b1, respectively; Appendix A). Notably, two genes, XM_021969502.1 and XM_021979071.1, were up-regulated more than 184-fold in response to all three inoculations (Figure 3B). XM_021969502.1 encoded an isoform of E3 ubiquitin-protein ligase PRT6, whereas XM_021979071.1 encoded an isoform of golgin subfamily A member 4 (Appendix A).

In the distal samples from A1M3 inoculation, a higher percentage of DEGs were up-regulated over 16-fold (Appendix A), suggesting a stronger response to the pathogen. In contrast, the percentages of DEGs up-regulated over 16-fold in the distal samples from A1M197 and 11116_b1 inoculations were smaller than those in the corresponding local samples, indicating different patterns of response to these three strains.

### 2.4. Functional Categories and GO Enrichment Analysis

The GO enrichment analysis revealed distinct regulation patterns in response to different Pss inoculations. The local samples from plants inoculated with the three Pss strains exhibited unique enrichment profiles; there was no overlap among the enrichment patterns of the three inoculations except for the term “Magnesium ion binding”, which was shared by A1M3 and 11116_b1 (Figure 4A–C). Specifically, in the samples from A1M3 inoculation, the most highly enriched GO terms were related to chloroplast structure and photosynthesis (Figure 4A); in the samples from A1M197 inoculation, the most highly enriched GO terms were associated with carbohydrate metabolism (Figure 4B); and in the samples from 11116_b1 inoculation, the most highly enriched GO terms were linked to terpene metabolism (Figure 4C). Notably, all the enriched DEGs in the three categories were down-regulated, indicating that infection with A1M3, A1M197, and 1111b_b1 could locally hinder the normal processes of photosynthesis, carbohydrate metabolism, and terpene metabolism, respectively, in sweet cherry. Additionally, A1M3 inoculation also caused the enrichment of DEGs encoding a group of transporters and oxidoreductase (Figure 4C), suggesting that this Pss strain might interfere with multiple aspects of cellular metabolism as well as the redox state of the cell.

The distal samples from plants inoculated with the three Pss strains showed more complicated enrichment patterns. On the one hand, the samples from A1M197 and 11116_b1 inoculations exhibited similarities in highly enriched GO terms, including “xyloglucan metabolic process”, “glucan metabolic process”, “cell wall polysaccharide metabolic process”, “cell wall biogenesis”, “hemicellulose metabolic process”, “cell wall macromolecule metabolic process”, and “xyloglucan: xyloglucosyl transferase activity” (Figure 4E,F). All the DEGs in these categories were up-regulated, suggesting that A1M197 and 11116_b1 infection could promote primary cell wall biogenesis in the distal tissues of sweet cherry. By contrast, in the samples from A1M3 inoculation, the highly enriched GO terms were related to photosynthesis (Figure 4D), and all the enriched DEGs in this category were down-regulated. This feature, together with Figure 4A, suggests that A1M3 infection could interfere with photosynthesis both locally and distally in sweet cherry. On the other hand, all three Pss strains caused the enrichment of the GO terms “terpene synthase activity” and “carbon-oxygen lyase activity” (Figure 4D–F), suggesting a certain extent of similar distal response triggered by the three strains. Most intriguingly, the GO term “defense response” was highly enriched in the samples from 11116_b1 inoculation, but not for the other two strains (Figure 4F). Also enriched for 11116_b1 was a group of GO terms related to ribonucleotide binding and transcription factor activity. This peculiar feature suggests that 11116_b1 could trigger defense mechanisms in the distal tissues of sweet cherry.

### 2.5. Validation of Expression Changes by Quantitative RT-PCR

To validate the expression profile of RNA-seq data, six genes were selected by comparing their transcript levels in samples infected with each Pss strain versus the Mock control using RNA-seq data (Appendix A). Specifically, three genes (GolginA-4, XM_021979071.1; E3 ubiquitin-protein ligase *PRT6*, XM_021969502.1; and CBS domain-containing protein, XM_021962599.1) were significantly up-regulated in local samples inoculated with each of the three Pss strains. Two genes (eIF-2-alpha kinase *GCN2*, XM_021957676.1 and GIGANTEA, XM_021962523.1) were significantly up-regulated in the distal samples infected with each Pss strain, and one gene (*AGD14*, XM_021952030.1) was significantly down-regulated in all the distal Pss-infected samples.

The expression of these genes was independently verified in the local and distal samples previously inoculated with each Pss strain by qPCR. In the local samples, GolginA-4 showed an increase in transcript level in samples infected with all three Pss strains compared to the mock control, but only the A1M3 strain-infected samples showed a significant increase (Figure 5A). Similarly, the transcript levels of E3 ubiquitin-protein ligase *PRT6* and CBS domain-containing protein were significantly increased in A1M3-infected samples (Figure 5B,C). In the distal samples, the selected gene eIF-2-alpha kinase *GCN2* presented increased transcript levels for all three Pss strains, whereas only A1M197- and A1M3-infected samples showed a significant increase compared to the mock condition (Figure 5D). GIGANTEA transcript levels were augmented, whereas *AGD14* was down-regulated in A1M3-infected samples (Figure 5E,F).

Interestingly, GolginA-4 and E3 ubiquitin-protein ligase *PRT6* also showed an increase in transcript level in the distal samples for all three Pss strains compared to the mock control. Although GolginA-4 levels were significantly increased in 11116_b1-infected samples, both transcript levels were increased in A1M3-infected samples (Figure 5G,H). However, the changes in the transcript levels of the genes GIGANTEA, GolginA-4, and E3 ubiquitin-protein ligase *PRT6* were less pronounced in the distal samples than in the local samples. Additionally, eIF-2-alpha kinase *GCN2* and GIGANTEA genes were up-regulated in the distal samples from all three Pss-infected twigs (Figure 5J,K). In the case of *AGD14*, down-regulation in A1M3-infected samples was detected as expected (Figure 5L).

Moreover, qPCR data from the selected genes and Actin7, a housekeeping gene, in local or distal infected samples were linearly, positively, and strongly correlated with RNA-seq data from the same genes (r = 0.7414; *p* < 0.0001; Appendix A). These results are consistent with the observed trend in RNA-seq data, validating the expression change analysis in local and distal Pss-infected conditions.

## 3. Discussion

### 3.1. Disease-Related Gene Candidates in Sweet Cherry Induced by Pss Inoculation

Heretofore, very little is understood about the molecular mechanism of defense responses in sweet cherry. Through conducting a comparative transcriptome analysis, a list of DEGs in response to Pss infection was identified, among which potential disease resistance genes were discovered. The GO enrichment analysis revealed that 33 DEGs caused by 11116_b1 inoculation were related to the defense response (Figure 4F), including XM_021974586.1, which encodes an RPM1-like protein; XM_021947717.1, which encodes an RPS6-like protein; XM_021977557.1, which encodes a TAO1-like protein; XM_021978626.1, which encodes an NDR1-like protein; XM_021961241.1, which encodes an SRC2-like protein; XM_021948923.1, which encodes an RPP13-like protein; XM_021976406.1, which encodes RGA3; XM_021976429.1, which encodes RGA4; XM_021969778.1, which encodes SNC1; XM_021977313.1, which encodes nematode resistance protein-like HSPRO2; XM_021953798.1, which encodes a homolog of probable disease resistance protein At5g66900 in *Arabidopsis*; XM_021946277.1, which encodes an uncharacterized protein; XM_021961188.1, which encodes NDR1/HIN1-like protein 3; XM_021961402.1 and XM_021961341.1, both of which encode BAP2-like proteins; and 18 genes encoding TMV resistance protein N-like proteins. With the exception of the uncharacterized protein, homologs of all these proteins have been studied in other plant systems [18,19,20,21,22,23,24,25,26,27,28,29], providing a starting point for elucidating the defense network in sweet cherry.

Interestingly, initial experiments on sweet cherry leaves revealed that 11116_b1 displayed the highest level of virulence when inoculated into leaf tissue (Appendix A). Therefore, this strain exhibited markedly distinct behaviors in local and distal tissues. In the local, inoculated tissues, it demonstrated the greatest virulence while triggering the fewest DEGs (Figure 2A). However, in the distal tissues, which were not in direct contact with the bacteria, it induced robust resistance responses, likely mediated through long-distance signal transduction pathways. All three Pss strains belong to subgroup 2d of *P. syringae* phylogroup 2; however, they produce different collections of type III secreted effectors (T3SEs) [30]. It can be speculated that the specific T3SEs of 11116_b1 contributed to the high virulence of this strain in the local tissue by suppressing plant resistance responses. On the other hand, signals specifically triggered by these effectors could travel to distal tissues, where they induced a cohort of resistance genes. Understanding the interaction between the T3SEs and the resistance genes would aid in protecting sweet cherry against pathogens. For instance, this knowledge could be utilized to trigger resistance genes and thwart Pss infection.

Notably, *TAO1* was also up-regulated 727- and 1390-fold in the local and distal samples from sweet cherry inoculated with A1M3 (Appendix A), despite no enrichment of defense-related DEGs being observed. TAO1 was found to contribute to disease resistance induced by AvrB, an avirulence factor of *P. syringae* pv. *tomato* DC3000, in *Arabidopsis* [26]. The up-regulation of *TAO1* by A1M3 suggests that this TIR-NBS-LRR disease resistance protein might play a role outside of the known defense pathway. Another noteworthy finding is that two genes encoding EDR2-like proteins, XM_021963532.1 and XM_021953322.1, were up-regulated 31- and 30-fold in distal samples from sweet cherry inoculated with A1M3 and 11116_b1, respectively. EDR2 is a negative regulator of salicylic acid-based defense in *Arabidopsis*, suppressing the hypersensitive response triggered by avirulent pathogens [31]. The up-regulation of *EDR2* by A1M3 infection also suggests its unknown function in the plant–pathogen interaction.

### 3.2. Pss Infection Triggers a Variety of Biological Processes

Monoterpenes, including α-pinene and β-pinene, have been reported to play multiple roles in plant defense responses [32,33]. Both are produced at the pathogen infection site and could activate the salicylic acid (SA) signaling pathway, initiating systemic acquired resistance (SAR). Additionally, both are volatile and could induce defense responses in neighboring plants. This study revealed the down-regulation of genes encoding a group of (-)-α-pinene synthase-like proteins in both local samples from 11116_b1 inoculation and distal samples from all three inoculations (Figure 4C–F, Appendix A). These results suggest that 11116_b1 could suppress the production of α-pinene at the infection site, inhibiting the activation of the SA signaling pathway and SAR. It may also block the interplant priming by volatility. Moreover, all three Pss strains could suppress α-pinene production in distal tissues, possibly blocking the systemic signal.

Chloroplasts play a pivotal role in plant defense by generating a burst of reactive oxygen species (ROS) upon the recognition of pathogen-associated molecular patterns (PAMPs), which triggers the basal immune program known as PAMP-triggered immunity (PTI) [34]. Successful pathogens may secrete effectors that reduce the production of ROS or enhance their scavenging, thereby suppressing the ROS burst and PTI [34,35]. This study showed that A1M3 infection caused the down-regulation of a large number of genes involved in chloroplast structure and photosynthesis (Figure 4A,C). Given that A1M3 is the least virulent strain among the three (Appendix A), it is likely that upon the infection of this strain, the regular function of chloroplasts was diverged to produce more ROS, leading to a ROS burst and subsequent immune response in sweet cherry, whereas the other two strains could suppress the ROS burst without down-regulating chloroplast-related genes. This speculation is supported by the enrichment of GO terms related to oxidoreductase activity in the local samples from A1M3 inoculation (Figure 4A).

One peculiar feature of the distal samples from A1M197 and 11116_b1 inoculations was the enrichment of GO terms related to primary cell wall biogenesis, especially hemicellulose metabolism (Figure 4E,F). The cell wall plays multiple roles in plant defense. At the infection site, it serves as a physical barrier against pathogens and as a source of signaling molecules [36]. On the other hand, systemic signals such as SA lead to the strengthening of the cell wall by depositing secondary components such as callose and lignin [37]. The altered metabolism of hemicellulose, a component of the primary cell wall, in the tissues distant from the infection site has not been previously reported. However, it could be speculated that certain signals triggered by the infection of A1M197 and 11116_b1 had travelled systemically to induce such a response in non-infected parts of the plant. Increased deposition of hemicellulose might strengthen the cell wall, albeit not as efficient as callose and lignin. It would be interesting to identify the signal and to study the function of hemicellulose in SAR.

Pathogen infection breaks the homeostasis of the host, forcing the plant to redistribute its resources to balance growth and defense [38]. To achieve the redistribution, contents of soluble carbohydrates such as glucose and sucrose are elevated, reflecting the up-regulation of related enzyme activity [39]. Interestingly, infection of A1M197 caused the down-regulation of genes involved in carbohydrate metabolism at the inoculation site, especially those encoding glucosyl hydrolases, glucosyltransferases, and sucrose synthases (Figure 4B, Appendix A), suggesting a reduced turnover rate of sucrose and other cellular carbohydrates. One possible explanation for this unexpected result might be that, at 40 days post-inoculation, with the establishment of bacterial population, carbohydrate synthesis was down-regulated to prevent the pathogen from exploiting its resources. This unique effect of A1M197, however, is worth exploring further.

### 3.3. Detection of Pss in Uninfected Samples

Using primers for the glutathione-dependent formaldehyde-activating enzyme 1 gene, Pss presence was detected in one mock-treated local sample and two distal samples (Appendix A). The primer pair was specifically designed to target a conserved sequence unique to Pss, ensuring that the positive result was not caused by the existence of other known *Pseudomonas* species or other *P. syringae* pathovars. The positive result, therefore, could be due to either experimental contamination of Pss or the existence of unknown *Pseudomonas* species. All the mock-treated twigs showed a healthy appearance at the time of sampling, and all the distal sections were kept intact and showed no symptoms at the time of sampling, indicating that Pss contamination, if it ever existed, did not cause infection and did not affect the results. On the other hand, there are *Pseudomonas* species existing in the environment and as endophytes [40] that might interfere with the detection Pss. Moreover, although most *P. syringae* strains are known as phytopathogens, some can exist in the environment without causing disease [41]. The Pss detected in the non-infected samples in this study could be a false positive result caused by other *Pseudomonas* species.

## 4. Materials and Methods

### 4.1. Plant Material and Growth Conditions

Two-year-old sweet cherry cv. ‘Lapins’ trees were purchased at a commercial plant nursery (34°28′12.0″ S 70°58′48.0″ W). These trees were transplanted to 20 L plastic pots filled with a mixture of peat moss, compost, and perlite (1:1:1, *v*/*v*/*v*) as substrate and supplemented with Basacote^®^ plus 6M (Combo expert, 3 g/L substrate) at dormant stage. Trees were acclimated into a side-open shade house placed in the experimental station at the Instituto de Ciencias Agroalimentarias Animales y Ambientales (ICA3) of the Universidad de O’Higgins, San Fernando, Chile (34°36′36.0″ S 70°59′24.0″ W). Each tree was irrigated twice per day by an automatic drip irrigation system until full pot capacity (approximately 0.85 L per irrigation event).

### 4.2. Bacterial Inoculation, Plant Sampling, Bacterial Re-Isolation, and Biochemical Characterization

Three Pss strains isolated from commercial sweet cherry orchards, A1M3, A1M197 [42], and 11116_b1 (this report), were grown for 20 h at 26 °C on KB agar medium, and the colonies were resuspended in sterile saline buffer (0.8% NaCl, 0.04% NaH_2_PO_4_ and 0.27% Na_2_HPO_4_), adjusted to 2 × 10^8^ CFU/mL (OD_600_ = 0.03). Six plants were inoculated per Pss strain, one twig per plant. Actively growing summer twigs were wounded at 10 cm from the apex with a sterile scalpel. The wound was immediately inoculated with 50 µL of one of the Pss suspensions or a sterile saline buffer as mock control, and immediately covered with sterile glycerol and Parafilm. Plants were maintained in the same shade house with the same irrigation as the acclimation period throughout the assay. Forty days after inoculation, twigs were cut at 30 cm from the apex, and disease symptoms were inspected around the wound. Green tissue beneath the epidermis was sampled (50–100 mg) next to the wound and 15–20 cm below the point of Pss inoculation for each inoculated twig. These samples were named local and distal samples, respectively.

Local samples were macerated in a sterile saline buffer, streaked on KB agar medium and incubated at 26 °C for 20 h. Two isolated colonies per sample were evaluated by LOPAT test [43] with minimum changes and by UV-induced fluorescence [17]. Oxidase assay was tested on oxidase strips (Sigma-Aldrich, Burlington, MA, USA) following the manufacturer’s instructions.

### 4.3. Nucleic Acid Sampling

Total RNA and DNA were extracted from the local and distal samples using the Spectrum™ Plant Total RNA Kit (Sigma) and FavorPrep™ Plant Genomic DNA Extraction Mini Kit (Favorgen, Wien, Austria), respectively. DNA and RNA quantity and RNA integrity were assessed with QUBIT 4 Fluorometer (Thermo Fisher Scientific, Waltham, MA, USA). For RNA samples, an IQ over 6 was considered acceptable in order to proceed sequencing.

### 4.4. Pss DNA Quantification in Sweet Cherry Samples

Calibration curves for DNA quantification were performed by qPCR using seven 10-fold serial dilutions of DNA extracted from A1M3 Pss strain cultivated in KB-Agar or total DNA from sweet cherry cv. ‘Lapins’ samples in an AriaMx Real-time PCR System. The qPCR reaction was performed using Brilliant III SYBR^®^ Green qPCR Master Mix (Agilent Technologies, Santa Clara, CA, USA) following the manufacturer’s instructions, with 200 nM of each primer and 1 μL of each dilution, supplemented with Molecular Biology Grade Water (Corning^®^, Corning, NY, USA) to reach the final volume of 15 µL. Glutathione-dependent formaldehyde-activating enzyme 1 gene primers were used to quantify the DNA of Pss, and SAR DEFICIENT 1 gene primers were used to quantify the DNA of sweet cherry in each sample (Appendix A). Each qPCR reaction was performed in duplicate. The obtained Pss calibration curve equation was: log_10_ [Pss DNA (ng/μL)] = (Cq − 13.6263)/(−3.302), *p*-value < 0.00001, r^2^ × 100 = 99.947%; the sweet cherry calibration curve formula was: log_10_ [sweet cherry DNA (ng/μL)] = (Cq − 20.7584)/(−2.50357), *p*-value = 0.0038, r^2^ × 100 = 99.243% (Appendix A). Pss DNA quantity in each sample was defined as the ratio between the DNA quantification from Pss and sweet cherry. Purified DNA from A1M3 Pss strain and purified DNA from an uninoculated *Prunus avium* twig were used as positive controls. Plant DNA extraction was diluted 20-fold with Molecular Biology Grade Water (Corning^®^) before qPCR reaction.

### 4.5. Library Construction and RNA Sequencing

RNA samples were treated with DNase and library was constructed using the TruSeq stranded total RNA with Ribo-Zero Plant Kit (Illumina, San Diego, CA, USA), which generated paired-end reads of 151 nt. Sequencing was performed using the Illumina platform. DNase treatment, library construction, and RNA sequencing were carried out by Psomagen (Rockville, MD, USA).

### 4.6. RNA Sequence Curation and Mapping to the Reference Transcriptome

The predicted transcripts from the protein-coding genes of *Prunus avium* cultivar ‘Satonishiki’ [12] were used as a reference to map the reads (GenBank GCA_002207925.1). The reference transcriptome contained 35,009 transcripts, corresponding to 25,841 predicted protein-coding genes. The raw data of RNA-seq were loaded onto the CLC Genomic Workbench software, version 21.0.5 (Qiagen, Hong Kong) for the following analyses. The raw reads were trimmed to remove the adapters and nucleotides of quality scores lower than 20. The trimmed reads from each individual sample were mapped to the reference transcriptome using the following parameters: similarity fraction = 0.95, length fraction = 0.8, insertion/deletion cost = 3, mismatch cost = 2, and unspecific match limit = 10. Principal component analysis (PCA) of all the datasets was conducted to validate the technical repeats of each treatment.

### 4.7. Differential Gene Expression Analysis

The differential expression analysis was performed with the CLC Genomic Workbench software by comparing the mapping results of the group of samples from the trees inoculated with each Pss strain to the mock-treated control group. The samples were normalized using the trimmed mean of M values (TMM) method [44]. A false discovery rate (FDR) of 0.05 was applied to the multiple sample testing. Differential expression between two sample sets was determined based on the transcripts with absolute fold change ≥ 2.0 and FDR-adjusted *p*-value ≤ 0.05. Heat maps were generated with the NG-CHM builder online (https://build.ngchm.net/NGCHM-web-builder/ (accessed on 28 September 2022)) [45,46].

### 4.8. Gene Ontology Enrichment Analysis

The lists of differentially expressed genes (DEGs) were entered into Database for Annotation, Visualization and Integrated Discovery (DAVID, https://david.ncifcrf.gov/home.jsp (accessed on 26 June 2023)) for gene ontology (GO) annotation with *p*-value ≤ 0.01 [47]. The enrichment bubble plots were generated by SRplot web server (http://www.bioinformatics.com.cn/srplot (accessed on 26 June 2023).

### 4.9. Validation of DEGs

RNA was treated with RNase-Free Dnase I (Thermo Fisher Scientific). cDNA was synthesized using an M-MLV Reverse Transcriptase (Invitrogen™, Waltham, MA, USA) with the oligo dT primer (5′-TTTTTTTTTTTTTTTTVN-3′) according to the manufacturer’s instructions. Six DEGs between mock samples and each Pss strain-inoculated sample were selected based on a significant and similar trend of changes in their expression (Appendix A). qPCR reactions were performed using Brilliant III Ultra-Fast SYBR Green qRT-PCR Master Mix (Agilent Technologies, CA, USA) in an AriaMx Real-time PCR System (Agilent Technologies, CA, USA) following the manufacturer’s instructions (5 ng of cDNA, 200 nM of each primer, 15 μL of total volume of reaction). Each qPCR reaction was performed in triplicate. Relative expression was calculated using the 2^−ΔΔCT^ with a primer efficiency correction [48]. Based on the RNA-seq data, two reference genes were used: peptidyl-prolyl cis-trans isomerase CYP21-1 gene and Splicing factor 3B subunit 1 gene (Appendix A). Simple linear regression analysis and coefficient determination between qPCR results and RNA-seq of the same genes were performed using Statgraphics Centurion XV, version 15.2.06.

## 5. Conclusions

The inoculation of *Pseudomonas syringae* pv. *syringae* strains induced differential gene expression in both local and distal tissues of sweet cherry cv. ‘Lapins’. The least virulent strain, A1M3, triggered the most extensive responses in local tissues, whereas the most virulent strain, 11116_b1, induced the strongest defense responses in distal tissues. DEGs involved in defense responses, terpene metabolism, photosynthesis, cell wall biogenesis, and carbohydrate metabolism were enriched in tissues from plants inoculated with different strains. These data serve as valuable groundwork for future research into sweet cherry defense mechanisms and the induction of immunity against Pss infection.

## Figures and Tables

**Figure 1 plants-12-03718-f001:**
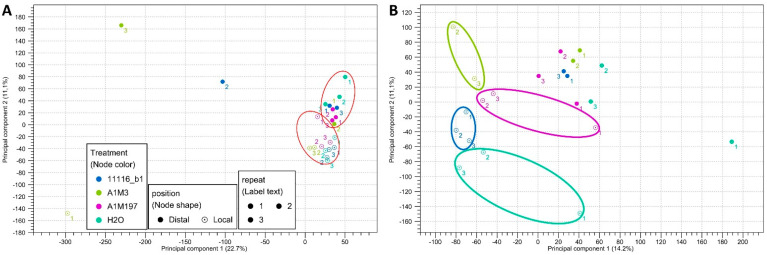
Principal component scatter plot (2D) of the transcriptome datasets from (**A**) the 24 samples of sweet cherry inoculated with three Pss strains and (**B**) the 21 samples circled in (**A**). The red circles in (**A**) indicate the clustering of local and distal datasets. The four circles in (**B**) indicate the clustering of datasets from different treatments.

**Figure 2 plants-12-03718-f002:**
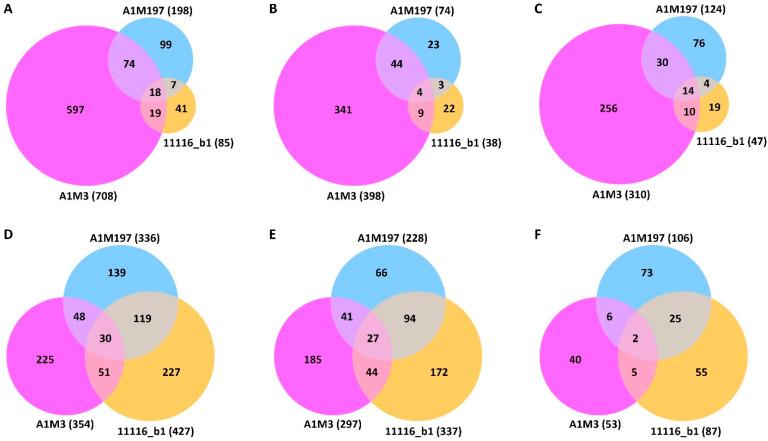
Venn diagrams showing differentially expressed sweet cherry genes post-inoculation of the three Pss strains. (**A**–**C**) Local samples. (**D**–**F**) Distal samples. (**A**,**D**) show the total numbers of DEG. (**B**,**E**) show the numbers of up-regulated genes. (**C**,**F**) show the numbers of down-regulated genes.

**Figure 3 plants-12-03718-f003:**
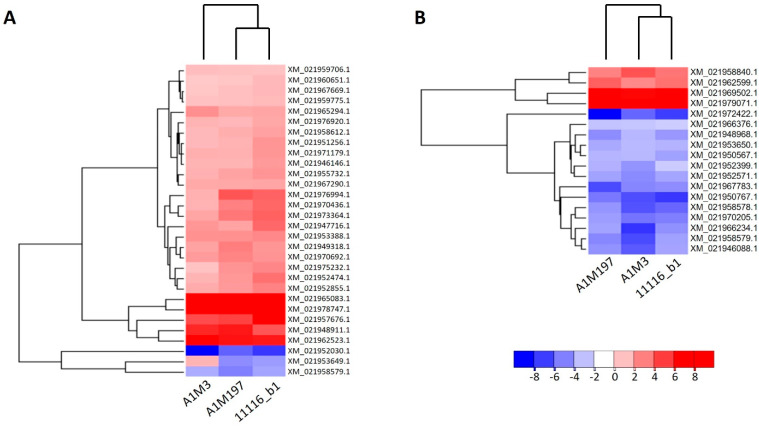
Heat maps showing the expression patterns of the DEGs after inoculation of all three Pss strains in sweet cherry. (**A**) Distal samples. (**B**) Local samples. Genes and datasets are clustered in hierarchical order according to their average Euclidean distances. The color gradient represents the log_2_ fold change in gene expression.

**Figure 4 plants-12-03718-f004:**
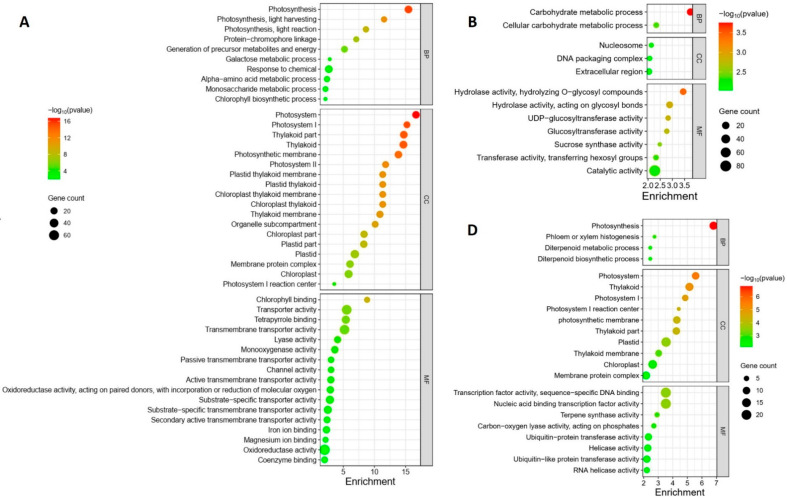
GO enrichment of DEGs in response to Pss inoculation. (**A**) Local samples inoculated with A1M3. (**B**) Local samples inoculated with A1M197. (**C**) Local samples inoculated with 11116_b1. (**D**) Distal samples inoculated with A1M3. (**E**) Distal samples inoculated with A1M197. (**F**) Distal samples inoculated with 11116_b1. Y axis, GO pathways. X axis, enrichment factors. BP, biological process. CC, cellular component. MF, molecular function.

**Figure 5 plants-12-03718-f005:**
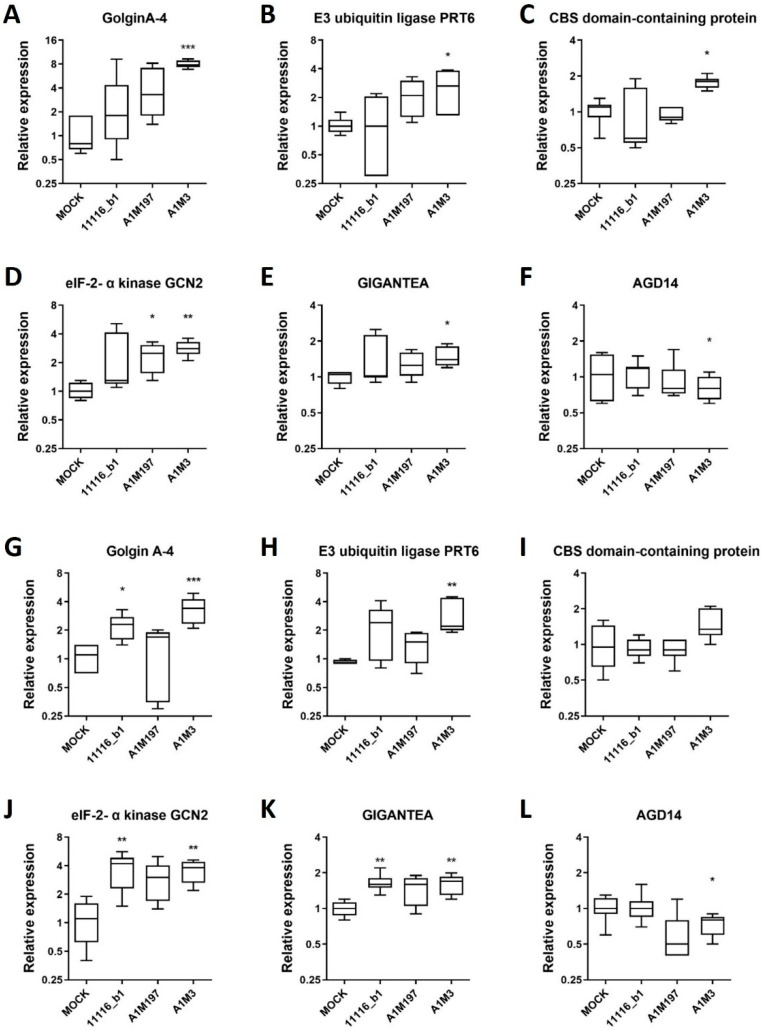
Expression profile of selected DEGs in local and distal samples from plants inoculated with Pss. Relative expression analysis of six genes from local (**A**–**F**) and distal (**G**–**L**) samples inoculated with 11116_b1, A1M197, and A1M3 Pss strains in sweet cherry twigs. The expression of these selected genes was measured by qPCR and normalized to the mock condition. Peptidyl-prolyl cis-trans isomerase *CYP21-1* and splicing factor 3B subunit 1 were used as reference genes. Kruskal–Wallis test with Dunn’s multiple comparisons test between inoculated and mock conditions was performed: ***, *p* < 0.0005; **, *p* < 0.005; *, *p* < 0.05; n = 3 (three independent biological replicates).

**Table 1 plants-12-03718-t001:** Statistics of the *Prunus avium* cv. Lapins transcriptome datasets.

Sampling Loci	Pss Strain	Replicates	Number of Raw Reads	Raw Read Length (nt)	GC Content (%)	Q20 (%)	Q30 (%)	Number of Reads after Trimming	Percentage of Trimmed Reads (%)	Average Read Length after Trimming (nt)	Number of Mapped Reads	Percentage of Mapped Reads (%)	Percentage of Transcripts with Mapped Reads (%)
Local	Mock	1	60,835,050	151	46.97	93.33	88.30	53,581,529	88.08	131.86	16,754,262	31.27	73.58
		2	56,187,144	151	47.78	94.38	89.62	50,721,363	90.27	130.61	16,331,122	32.20	73.87
		3	63,132,360	151	47.13	95.93	91.59	58,677,218	92.94	131.63	18,513,496	31.55	75.29
	A1M3	1	63,225,668	151	51.82	93.10	87.88	55,555,633	87.87	126.43	3,691,298	6.64	59.78
		2	56,813,968	151	46.17	96.77	92.50	54,199,081	95.40	133.34	20,805,818	38.39	74.33
		3	57,510,590	151	49.47	94.32	89.64	51,953,024	90.34	131.69	13,244,942	25.49	71.46
	11116_b1	1	57,931,388	151	45.78	93.15	87.95	51,048,941	88.12	133.41	18,382,040	36.01	74.84
		2	53,399,388	151	46.72	95.43	90.92	49,360,607	92.44	131.64	17,792,020	36.04	74.85
		3	55,546,824	151	47.12	95.68	91.31	51,399,548	92.53	132.19	18,189,231	35.39	73.67
	A1M197	1	69,961,046	151	47.66	96.24	91.91	66,028,852	94.38	131.24	20,000,527	30.29	73.68
		2	58,834,920	151	46.78	92.03	86.79	50,531,708	85.89	132.67	19,298,925	38.19	73.88
		3	57,500,624	151	45.74	96.28	91.95	54,062,414	94.02	134.25	21,404,629	39.59	74.23
Distal	Mock	1	61,368,428	151	45.17	94.02	88.89	54,821,582	89.33	132.26	16,397,417	29.91	69.54
		2	60,884,042	151	45.26	93.50	88.49	53,950,576	88.61	132.14	18,244,099	33.82	72.18
		3	60,044,718	151	47.31	94.40	89.65	54,180,939	90.23	131.93	14,021,993	25.88	71.25
	A1M3	1	60,741,374	151	46.02	93.30	88.19	53,603,105	88.25	132.87	16,711,876	31.18	71.91
		2	53,215,158	151	45.80	93.35	88.33	46,916,794	88.16	133.11	17,700,614	37.73	75.20
		3	55,417,944	151	53.13	93.98	89.09	49,778,523	89.82	126.38	4,567,419	9.18	59.66
	11116_b1	1	62,375,428	151	46.85	94.28	89.55	56,243,457	90.17	130.01	17,689,918	31.45	71.68
		2	56,401,534	151	51.33	93.98	89.12	50,643,914	89.79	126.47	10,970,689	21.66	67.01
		3	56,548,966	151	46.17	95.62	91.40	52,320,030	92.52	132.07	17,770,745	33.97	72.08
	A1M197	1	53,984,686	151	46.55	94.71	90.03	49,040,902	90.84	131.09	14,992,085	30.57	71.26
		2	54,615,978	151	45.94	93.61	88.62	48,392,367	88.60	133.17	14,639,217	30.25	70.58
		3	60,384,644	151	45.74	94.44	89.64	54,514,976	90.28	133.59	17,940,683	32.91	73.16

## Data Availability

The RNA-Seq data was deposited in the sequence read archive of the National Center for Biotechnology Information under accession number PRJNA1014457.

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
