# Peer review of "Transcriptome Analysis of Sweet Cherry (Prunus avium L.) Cultivar ‘Lapins’ upon Infection of Pseudomonas syringae pv. syringae"

_plants, 2023, doi:10.3390/plants12213718_

Round 1

Reviewer 1 Report

Cai et al. perform a transcriptome analysis of sweet cherry Prunus avium inoculated with the cherry canker pathogen Pseudomonas syringae. This is a compatible reaction so they are measuring the response of a susceptible plant to this pathogen. They identify various genes that are differentially expressed in local vs. Distal tissues, genes that vary in their response to pathogen strains and importantly some genes involved in defense responses that show differential expression. These findings are useful for those researchers interested in the molecular interaction of bacteria with woody plants. The article in general is descriptive and does not validate the role of any of these gene expression changes in this interaction, however it is a useful dataset.

A major comment to be addressed is that the Pseudomonas strains are not characterized so we do not know about them - if they are closely related and what their virulence levels are. Pss is a very diverse grouping of the P. syringae species complex and can be divided into several subgroups 2a, 2b, 2d (almost analogous to separate species), do you know if these strains are in the same subgroup? Different subgroups might explain the differences in plant responses. It would also significantly improve the manuscript if pathogenicity data for these strains could be shown. Seeing as this research primarily focuses on the interaction of the Pseudomonas strains with their host I do not believe that it is OK to quote "unpublished data" as on page 15 and 16. Many of the results involve comparing the responses to these strains but we do not know anything about them in terms of their pathogenicity and relationship to each other. The authors should add pathogenicity and genotyping data of these strains to the manuscript.

It is mentioned on page 15 that strain 11116_b1 is most virulent and has the most effectors but this data is not shown? Again I do not agree with the quotation of unpublished data which is so relevant to the results of the current study. Also it is stated that it is more virulent but induces a greater resistance response - this sentence does not make sense? Surely if it is more virulent it would better suppress the immune response.

Have the sequencing reads been deposited in genbank SRA or similar repository? This should be done before publication. 

Explain how you decided what was local vs. distal distance for sampling? Was this based on previous observations of the lesion lengths. Did any lesions reach the distal tissue?

Specific comments are below:

1.    P2 paragraph 1: P. syringae pv. unknown (Omrani et al., 2019)? Why is it unknown, is it not just Pss?

2.    P2 last paragraph: What was the optical density of the Pss at 2x10^8?

3.    P2 last paragraph: By seasonal twigs do you mean the actively growing tissue? What time of year was the inoculation done was the tissue actively growing or dormant?

4.     P3 3rd paragraph: Was the RNA treated with DNase before library preparation?

The English language is fine 

Reviewer 2 Report

Comments to the Author:

Title: Transcriptome analysis of sweet cherry (Prunus avium L.) cultivar ‘Lapins’ upon infection of Pseudomonas syringae pv. syringae

Overview and general recommendation:

The manuscript deals with an important topic related to the Transcriptome analysis of sweet cherry (Prunus avium L.) cultivar ‘Lapins’ upon infection of Pseudomonas syringae pv. syringae. The manuscript technically sounds well and shows high novelty. It is well written and shows the need for minor linguistic adjustments. In this regard, the needed adjustments are highlighted in “Minor comments” section. Also, numerous statements lack reliable sources (references) that should be provided. Adding to that, the first voice form of sentences shall be avoided in a scientific paper and the impersonal form shall be adopted instead.

The Abstract section outlines clearly the problematic, aims, methodology and findings of the current study while reporting the main conclusions aroused. The Introduction section is well structured and aiming and underlines appropriately the whole subject under study. The aims of the study are also clear and understood. The Materials and methods section is clear, well written, and encloses all the information related to the adopted methodology, and statistical analysis. The Results section shows a correct representation of the findings along with an appropriate scientific analysis. The Discussion section shows a correct comparison between the findings of this study and earlier published ones in literature. An appropriate Conclusions section was added in which authors briefly summarized the findings of their study. However, they shall suggest further related research being based on the raised assumptions.

My comments and queries for authors are detailed below in “Minor comments” section.

1.1.            Minor comments:

1-      The in-text references presentation shall be adjusted following the journal’s guidelines.

2-      Abstract, Page 1: “In order… inoculation”: Kindly avoid the first voice form of the sentence and adopt the impersonal form instead.

3-      Abstract, Page 1: “Also… biogenesis”: The sentence is cumbersome; accordingly, kindly reformulate in order to make it clearer and more aiming.

4-      Abstract, Page 1: Kindly adjust as follow: “This study”.

5-      Keywords, Page 1: Kindly adjust as follow: “local response” and “distal response”.

6-      1. Introduction, Page 2: “In this study… (Pss)”: Kindly avoid the first voice form of the sentence and adopt the impersonal form instead.

7-      1. Introduction, Page 2: “Three Pss… three strains”: The sentence is long and cumbersome; accordingly, kindly reformulate in order to make it more concise, clearer and more aiming.

8-      2. Materials and Methods, 2.1. Plant material and growth conditions, Page 2: Kindly mention the number of trees used in this study.

9-      2. Materials and Methods, 2.1. Plant material and growth conditions, Page 2: Kindly replace “plants” by “trees”.

10-  2. Materials and Methods, 2.1. Plant material and growth conditions, Page 2: At which growing stage was the supplementation performed? Kindly mention.

11-  2. Materials and Methods, 2.2. Bacterial inoculation, plant sampling, bacterial re-isolation, and biochemical characterization, Pages 2–3: Kindly put a space between the number and the temperature’s unit.

12-  2. Materials and Methods, 2.6. RNA sequence curation and mapping to the reference transcriptome, Page 3: Kindly adjust as follow: “contained”.

13-  3. Results, 3.1. Disease development, Page 4: “To confirm… near the wound”: Kindly avoid the first voice form of the sentence and adopt the impersonal form instead.

14-  3. Results, 3.3. Different Pss strains trigger different local and distal responses, Page 8: Kindly adjust as follow: “were comparable”.

15-  3. Results, 3.3. Different Pss strains trigger different local and distal responses, Page 8: Kindly adjust as follow: “two to six times higher than the latter”.

16-  3. Results, 3.3. Different Pss strains trigger different local and distal responses, Page 8: “Four DEGs… 11116_b1”: The sentence is long and cumbersome; accordingly, kindly reformulate in order to make it more concise, clearer and more aiming.

17-  3. Results, 3.3. Different Pss strains trigger different local and distal responses, Page 9: “The fold change… significantly lower”: Same recommendation as in the previous comment.

18-  3. Results, 3.3. Different Pss strains trigger different local and distal responses, Page 9: Kindly adjust as follow: “encoded”.

19-  3. Results, 3.4. Functional categories and GO enrichment analysis, Page 12: “Most intriguingly… (Figure 4F)”: The sentence is cumbersome; accordingly, kindly reformulate in order to make it clearer and more aiming.

20-  4. Discussion, 4.1. Disease-related gene candidates in sweet cherry induced by Pss inoculation, Page 15: “Through conducting… were discovered”: Kindly avoid the first voice form of the sentence and adopt the impersonal form instead.

21-  4. Discussion, 4.2. Pss infection triggers a variety of biological processes, Page 16: “In this study… three inoculations”: Same recommendation as in the previous comment.

22-  4. Discussion, 4.2. Pss infection triggers a variety of biological processes, Page 16: “Chloroplasts… (PTI)”: This statement lacks reliable sources (references); accordingly, kindly provide them.

23-  4. Discussion, 4.2. Pss infection triggers a variety of biological processes, Page 16: “In this study… photosynthesis”: Kindly avoid the first voice form of the sentence and adopt the impersonal form instead.

24-  4. Discussion, 4.2. Pss infection triggers a variety of biological processes, Page 16: “Pathogen infection… and defense”: This statement lacks reliable sources (references); accordingly, kindly provide them.

25-  4. Discussion, 4.2. Pss infection triggers a variety of biological processes, Page 16: Kindly adjust as follow: “post-inoculation”.

26-  5. Conclusions, Page 17: Kindly add a sentence at the end of this section in which you suggest further related research being based on the raised assumptions from the current study.

The manuscript is well written and shows the need for minor linguistic adjustments.

Round 2

Reviewer 1 Report

The authors have addressed all my major comments. On re-checking the figures, it is clear from the PCA plot that infection did not seem to make a major impact on the transcriptome. The authors should add some description of what the PCA plot means to their results. Wouldn't you expect there to be separate clusters for the different treatments on the PCA plot? Did you find different clusters appeared once you removed the outliers. As different treatments (infected vs. not) overlap what this means for the subsequent analysis should be discussed.

From inspection of the PCA it appears the distal and local may be forming different clusters this should be discussed. Currently with the three outliers included any pattern is difficult to both see and read. 

This should be addressed before it is ready for publication. 
